# Dynamical analysis of financial stocks network: Improving forecasting using network properties

Ixandra Achitouv [ORCID]*

Institut des Systèmes Complexes ISC-PIF, CNRS, Paris, France

* ixandra.achitouv@cnrs.fr

## Abstract

Applying a network analysis to stock return correlations, we study the dynamical properties of the network and how they correlate with the market return, finding meaningful variables that partially capture the complex dynamical processes of stock interactions and the market structure. We then use the individual properties of stocks within the network along with the global ones, to find correlations with the future returns of individual S&P 500 stocks. Applying these properties as input variables for forecasting, we find a 21% improvement on the R2score in the prediction of stock returns on long time scales (per year), and 3% on short time scales (2 days), relative to baseline models without network variables. These findings highlight the potential of integrating network-based variables into stock return prediction models, which could enhance forecasting accuracy and provide a deeper understanding of market dynamics. This approach could be valuable for both investors and researchers seeking to model and predict stock behaviour in complex financial networks.

## Introduction

Financial market dynamics exhibit properties of complex systems, such as non-linear behaviour and inter-dependencies of stock prices [1–3], emergent phenomena such as trends and bubbles from interactions of market participants [4–6], adaptive behaviour [7,8] and feedback mechanisms [9] (e.g. rising prices attract more investors, further driving up prices). Collective behaviours emerging in financial markets are known as market modes, referring to synchronized movements of groups of stocks. They have been studied in stock return correlations using different tools including principal component analysis (e.g. [10]), complex network analysis (e.g. [1,11–13]) and random matrix theory (e.g. [14–16]). Indeed, in a complex system, the behaviour of individual components, or agents, is coupled to the collective dynamics of the system. When applied to financial markets, one could ask how these couplings can be extracted and used to improve forecasting of the global stock market as well as in individual stock returns.

Many studies have proposed to forecast individual stock returns using soft computing methods [17,18,18] (e.g. machine learning regression algorithms) using mostly individual features of a stock (e.g. previous closing value, difference between high and low value of the stock

**Data availability statement:** All relevant data for this study are publicly available from the figshare repository (https://doi.org/10.6084/m9.figshare.28238414.v1).

**Funding:** The author(s) received no specific funding for this work.

each day, mean volume exchanged, etc..). Others have included macroeconomic variables to forecast stock returns (e.g. [19]), or sentiment analysis using natural language processing techniques (e.g. [20–22]). Some studies have analyzed stock market networks' structural properties to detect influencers in stock market "communities" (e.g. [23,24]) but little research has been conducted to apply the properties of a network built on stock returns correlations to forecast the future movements of the global stock market [25,26], or of individual stocks [27]. Unlike previous works that often focuses on static network structures or market-wide influence detection, this study applies dynamic network analysis to understand how stock return interdependencies evolve over time, offering insights into the collective behaviours of the market. Previous methodologies, such as [26], focused on applying network properties to predict market movements but did not investigate how these properties change over time and can influence individual stock forecasts.

In this article, we use complex network analysis to study the dynamical inter-dependencies of stock returns and their collective behaviours, and test how the properties of the network correlate with the market stock return that we define as the unweighted average return across all assets. Indeed, networks underpin complex systems and influence their behaviour [28]. Therefore, by analysing the dynamical evolution of the stock returns network one could hope to infer the overall dynamics of the stock returns. We further consider the properties of individual stocks within the network and test how these properties correlate with the future returns of individual stocks. Finally, we consider how our results change depending on the time scale employed to built the network, to test if we have a scale invariance on the correlations observed between the log return and the network properties. Indeed, scale-invariance is often considered a property of complex systems [29], including in financial market [29,30]. The fractal market hypothesis, discussed in [31–34] suggests that market prices exhibit fractal properties and that patterns observed in short-term price movements can resemble those in long-term trends. This contrasts with the efficient market hypothesis, which assumes that market prices follow a random walk and are inherently unpredictable. Thus it becomes interesting to test the added value of the network input variable on short and long time scale forecasting and study if the key predictors are the same. Thus, this work extends previous studies in financial network analysis by explicitly considering the scale-invariance of network correlations over different time scales and testing how network features serve as reliable predictors for future stock returns.

This article is organized as follows: first, we describe the key network properties we consider and how we constructed our financial network. Second, we measure the dynamic evolution of these properties and test their correlation with the overall market return. Third, we examine the individual properties of the stocks within the network and test if they have meaningful correlations with their future returns. We then use these properties to forecast individual stock returns. Finally, we present our conclusion.

## Network properties and construction of the financial stock return network

### Network properties

In order to study the properties of the network we consider centrality measures:
(1) The degree $k_i$ of a node $i$ in a network represents the number of connections it has to other nodes [28]. In an undirected network, the degree $k_i$ of a node $i$ is given by the sum of the adjacency matrix elements corresponding to that node.

$$k_i = \sum_j a_{ij} \tag{1}$$

where $a_{ij}$ is an element of the adjacency matrix A, which is 1 if there is an edge between nodes $i$ and $j$, and 0 otherwise.

(2) Closeness centrality $C_C(u)$ of a node $u$ measures the average length of the shortest path from a node to all other nodes in the network, representing how quickly information spreads from a given node to others [35]. It identifies nodes that can quickly interact with all other nodes, and is defined as:

$$C_C(u) = \frac{1}{\sum_{v \in V} d(u,v)}$$ (2)

where $d(u,v)$ is the shortest path distance between nodes $u$ and $v$, and V is the set of all nodes in the network.

(3) Betweenness centrality $C_B(v)$ of a node $v$ is a measure of the extent to which a node lies on the shortest paths between other nodes, indicating its role as a bridge or connector within the network. Nodes with high betweenness centrality can control the flow of information or resources between different parts of the network [36].

$$C_B(v) = \sum_{v \neq s \neq t} \frac{\sigma_{st}(v)}{\sigma_{st}}$$ (3)

where $\sigma_{st}$ is the total number of shortest paths from node $s$ to node $t$ and $\sigma_{st}(v)$ is the number of those paths that pass through $v$.

(4) Eigenvector centrality $C_E(i)$ of a node $i$ is a measure of the influence of a node in a network based on the idea that connections to high-scoring nodes contribute more to the score of the node in question [37]. A high eigenvector score means that a node is connected to many nodes who themselves have high scores. It is computed using the principal eigenvector of the adjacency matrix of the network:

$$C_E(i) = \frac{1}{\lambda} \sum_{j \in \mathcal{N}(i)} A_{ij} C_E(j)$$ (4)

where $\lambda$ is a constant (the eigenvalue), $\mathcal{N}(i)$ is the set of neighbors of $i$, and A is the adjacency matrix of the network. Applied to a financial market, high eigenvector values can be problematic for systemic risks as contagion risk is linked to these influential stocks [38]. Indeed, a highly interconnected financial network can act as an amplification mechanism by creating channels for a shock to spread, leading to losses that are much larger than the initial changes [39].

In addition, we consider the weighted clustering coefficient $C_C(i)$ of a node $i$, it measures the tendency of nodes to cluster together while taking into account the strength of the connections (weights) between nodes, providing a more nuanced view of network cohesiveness [40]:

$$C_C(i) = \frac{1}{s_i(k_i - 1)} \sum_{j,h} \frac{(w_{ij} + w_{ih})}{2} a_{ij} a_{ih} a_{jh}$$ (5)

where $s_i$ is the sum of the weights of the edges connected to node $i$, $k_i$ is the degree of node $i$, $w_{ij}$ is the weight of the edge between nodes $i$ and $j$, and $a_{ij}$ is the adjacency matrix element that is 1 if nodes $i$ and $j$ are connected and 0 otherwise. Applied to a financial market, networks with high clustering coefficients are often more robust and resilient to node failures, as the redundancy in connections within clusters can help maintain the overall connectivity [41]. This could be used as a network systemic risk measure similarly to the eigenvector centrality,

when building on optimal portfolio. In fact it was found (e.g. [42]) that one can improve on traditional portfolio selection by maximizing the usual Sharpe ratio where the standard deviation of the portfolio is replaced by a weighted average of node centrality measures (e.g. eigenvector centrality, betweennness centrality, closeness centrality).

We also measure properties that characterize the network structure globally, such as:

(1) Community stability measure. In a network, this refers to the persistence of community structures over time or under perturbations, indicating how robust the communities are to changes in the network. It can be quantitatively assessed using various metrics, one of which involves evaluating the modularity Q of the network [43]. The modularity of a network is defined as:

$$Q = \frac{1}{2m} \sum_{ij} (A_{ij} - \frac{k_i k_j}{2m}) \delta(c_i, c_j) \tag{6}$$

where $A_{ij}$ is the adjacency matrix, $k_i$ and $k_j$ are the degrees of nodes $i$ and $j$ respectively, $m$ is the total number of edges, and $\delta(c_i, c_j)$ is a function that is 1 if nodes $i$ and $j$ are in the same community and 0 otherwise.

(2) The largest component L, defined as the largest subset of nodes such that there is a path connecting any two nodes within this subset. It represents the most extensive cluster of interconnected nodes in the network, providing insights into the network's structure and functionality [28]. Let G=(V,E) be a graph where V is the set of vertices and E is the set of edges. If $C_i$ represents the $i$-th connected component of G, then the largest component L is given by:

$$L = max_i \, | \, C_i \, | \tag{7}$$

where $| \, C_i \, |$ is the size (number of nodes) of component $C_i$.

(3) The resilience of a network, R, measures its ability to maintain its structural integrity and functionality in the face of failures or attacks on its nodes or edges. It characterizes the robustness and vulnerability of networks [28,41]. Resilience can be quantified by assessing the size of the largest connected component after a certain fraction of nodes or edges have been removed.

$$R(f) = \frac{| \, L(f) \, |}{| \, V \, |} \tag{8}$$

where $| \, L(f) \, |$ is the size of the largest connected component after removing $f | \, V \, |$ nodes, and $| \, V \, |$ is the total number of nodes in the original network.

## Construction of the network

The S&P 500 index was chosen for this study as it represents a diverse cross-section of the largest U.S. companies across various sectors, making it a widely recognized benchmark for the overall stock market and an ideal candidate for studying network properties and their correlation with market dynamics. The closing stock prices can be downloaded from *Yahoo* finance. The periods chosen for this analysis were selected to balance the availability of comprehensive data with the need to capture meaningful long-term and short-term market dynamics. For the long-term period, we selected daily stock closing prices spanning from 1993-01-01 to 2024-01-01 to ensure coverage of multiple economic cycles, including major market events such as the dot-com bubble, the 2008 financial crisis, and the COVID-19 pandemic, which provide a robust basis for analyzing the dynamics of stock return networks over extended time scales. For the short-term period, the data from 2022-08-31 to 2024-07-31 captures recent market activity, offering insights into network properties under current market

conditions while focusing on higher-frequency (hourly) dynamics. Both datasets used for this analysis can be accessed on https://github.com/IxandraAchitouv/Dyn_analysis_FSN_forecasting. The acronyms of the stocks from 1993 to 2024 can be accessed in the same link under the name historicalsp500components.csv.

For the long term period,

we clear out stocks that were not present in the entire time range, ending up with 267 stocks and 7805 days of closing values for each stock. We then split our data into 30 yearly samples covering *Nobs* = 30 years of business days.

For the short term period,

we also clear out stocks that were not present in the entire time range, ending up with 488 stocks and 3346 hours of recorded price for each stock. We then split our data in time intervals of 14 hours (corresponding to two business days) and we end up with *Nobs* =238 measures (one measure every two business days).

The structure of the financial market can be mapped as a network where nodes represent the stocks and the edges connecting them represent the correlations between their returns after applying some filtering, e.g. [1,44–46].

To compute the correlation of their returns we take the closing price $P_i(t)$ log-return:

$$C_{i,j} = \frac{<r_i(t)r_j(t)> - <r_i(t)><r_j(t)>}{\sigma_i\sigma_j} \tag{9}$$

where $r_i(t) = \log[P_i(t)] - \log[P_i(t-1)]$ and $\sigma_i$ is the standard deviation of the stock computed over the year we consider.

In this correlation matrix all stocks are connected to one another (the degree distribution is therefore a constant equal to the number of nodes), which does not provide any useful information. One approach to convert this correlation matrix to an adjacency matrix $A_{i,j}$, is the threshold method originally introduced in [44] and used in many studies e.g. [23,24,47]. It is defined as $A_{i,j} = C_{i,j}$ if $|C_{i,j}| \geq \rho_c$, otherwise $A_{i,j} = 0$. One can easily understand what it does qualitatively: it filters out 'spurious' correlations and keeps an edge between stocks when the correlation of their return is above a certain limit. Thus remains the question of which value one should take for the threshold to optimally filter the correlation matrix of the returns, differentiating between noise and signal. Note that one could start by extracting the market modes from the correlation matrix as in [13] before applying the threshold, however this requires the correlation matrix to be full rank which may not be the case for short time scales. In previous studies, the value for the threshold is usually an heuristic choice as discussed in [48]. For instance, in [49], $\rho_c = <C_{i,j}> + n\sigma$, where $\sigma$ is the standard deviation of the $C_{i,j}$ distribution and $n$ an integer. Recently, a criterion based on network properties was introduce in [50]. In our work we also propose to privilege a value based on the network. Many complex systems naturally evolve towards a scale-free structure due to the dynamic processes governing their formation. The scale-free property means that the degree distribution follows a power law: a few nodes are connected with many (high degree) while most are not connected (null degree). The scale-free network emerges because the probability of a node gaining new connections is proportional to its existing connectivity [51]. In fact, many natural networks grow over time by the addition of new nodes. New nodes are more likely to attach to existing nodes that already have a high degree of connectivity, leading to a few highly connected hubs. This process, known as preferential attachment, is a fundamental mechanism in the formation of scale-free networks. Scale-free networks may also provide an evolutionary advantage by enabling complex systems to adapt and evolve more efficiently. Highly connected

nodes (hubs) can serve as control points that help the network respond to changes and perturbations, thus enhancing the system's overall adaptability and resilience [52]. Hence we propose to define the threshold value as the minimum value where the degree distribution of the nodes follows a power law.

In practice, we implement a loop starting with an initial threshold value of $\rho_c = 0.8$. For each iteration, we compute the adjacency matrix and the degree distribution of the nodes. If the degree distribution is not convex, we increment the threshold by 0.1 and repeat the process. This iterative procedure continues until the degree distribution becomes convex, at which point we stop, resulting in threshold values of $\rho_c \sim 0.9$. This threshold can be seen as a hyperparameter for the construction of the adjacency matrix [23]. While exploring alternative criteria for threshold selection goes beyond the scope of this analysis, we note that as long as the threshold value is sufficiently large to filter meaningful correlations, the results should remain robust.

Once we have built the adjacency matrix, we use the *NetworkX* library [53] to visualize the network and compute most of the network statistics we previously introduced.

In Fig 1 we display the resulting networks for 4 different years (top panels), where nodes are colored by the sector of the stocks. The spatial visualisation of the network is computed using ForceAtlas2 [54] which maximizes/minimizes the distance of nodes that have low/high weighted edges respectively. The size of the nodes is proportional to their degree. Interestingly we can observe some clustering where stocks of the same sector (shown by colors) are closer. We run a Louvain community finder [55] which optimizes locally the difference between the number of edges between nodes in a community and the expected number of such edges in a random graph with the same degree sequence. As a result, we identify different numbers of clusters depending on the year (Ncluster). For visibility we display the label of the nodes if the node is in the top 3% of the eigenvector centrality distribution, when its eigenvector centrality is the maximum within the Ncluster, or when it has the largest eigenvector centrality within its S&P5500 sector.

In the lower panels of Fig 1 we display the degree distribution of the nodes as well as the histograms of eigenvector centrality and local clustering. The red vertical line corresponds to the mean. In the appendix we also show 4 selected random networks built using the short time period.

## Global evolution of the network and the market stock returns

To characterize the global dynamical evolution of the network we consider the following measures: the average degree of the top 90% nodes, the mean closeness centrality (the mean corresponds to the average over all nodes), the mean betweeness centrality, the mean eigenvector centrality, the mean clustering, the largest component, the resilience of the network, and the community stability. We also measure the maximum eigenvalues of stock returns computed from their correlation matrix, as this can used as a precursor to financial crises by indicating market turbulence and systemic risk [14]. The global dynamical properties of the graphs for the long and the short term periods can be accessed at https://github.com/IxandraAchitouv/Dyn_analysis_FSN_forecasting.

### Long term period

In Fig 2 we display these measurements (black curves) along with the global stock returns evolution (blue curves) rescaled by some constant factor to fit on the same y-axes.

The vertical dashed lines correspond to (from left to right): The Asian Financial Crisis (1997), which primarily affected East Asian markets but also had a global impact. The U.S.

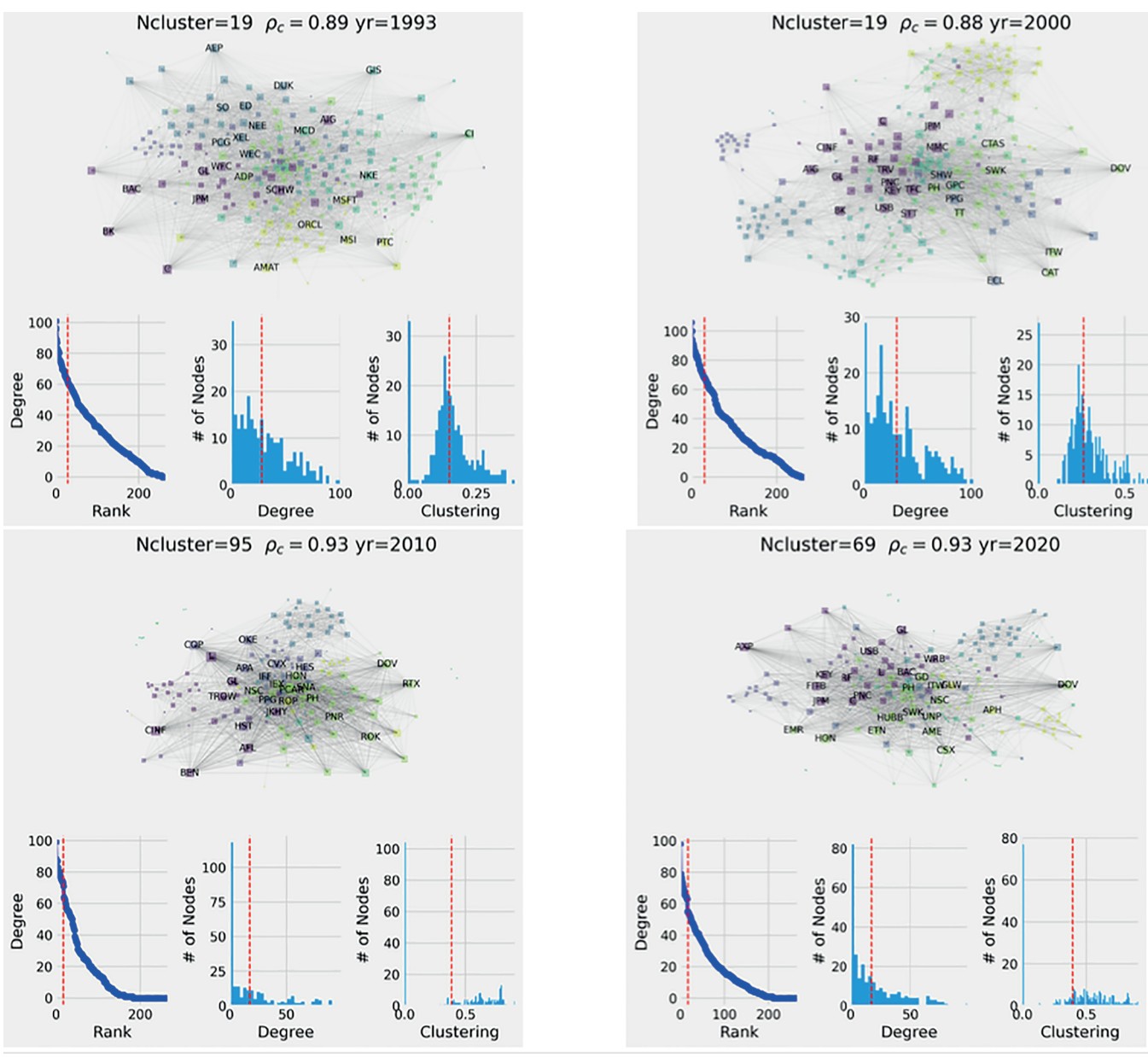

**Fig 1. Network of the stocks for different years and their properties for the long time scale.**

market experienced a decline in 1998 due to global instability. - The Dotcom Bubble Burst (2000–2002) led to a sharp decline in the S&P 500 as technology stocks crashed. - The Subprime Crisis (2008), triggered by the subprime mortgage meltdown and global banking failures, caused a significant plunge in the S&P 500. - The Federal Reserve announced its third round of quantitative easing (QE3) in September 2012, committing to purchase 40 billion USD in mortgage-backed securities per month to boost economic growth and reduce unemployment. The S&P 500 responded positively, gaining momentum in the latter half of 2012. - The COVID-19 pandemic caused severe market disruptions, leading to a sharp and rapid decline in the S&P 500 in March 2020. However, the market rebounded quickly after the

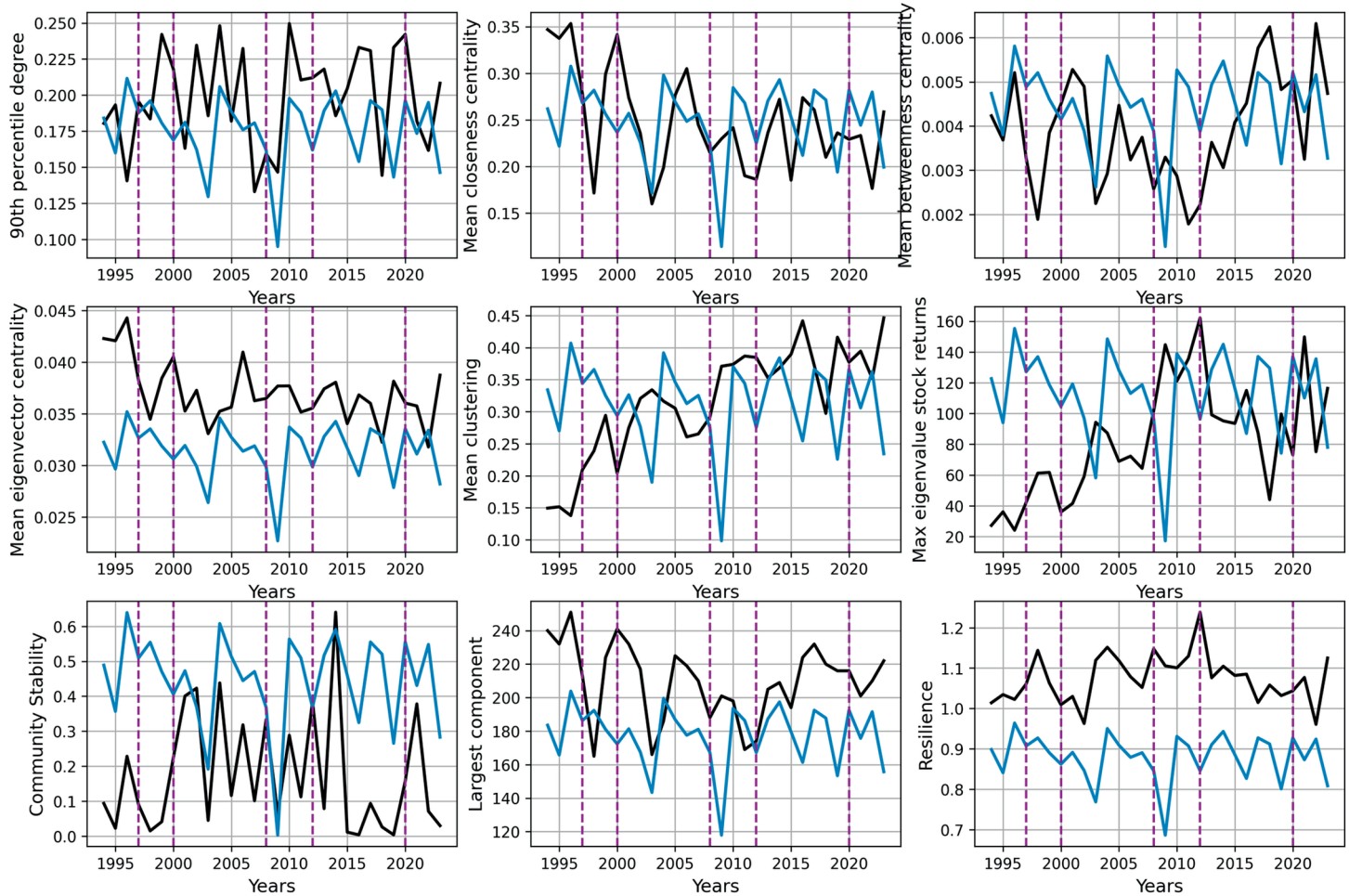

**Fig 2. Evolution of the network properties as a function of the year (black curves) and the stock return rescaled (blue curves).** The number of observed time scale is *Nobs* = 30. The vertical dashed lines correspond to (from left to right): Asian Financial Crisis, 1997 - Dotcom Crash, 2000 - Subprime Crisis, 2008 - Federal Reserve's QE3 Announcement, 2012 - COVID-19 pandemic, 2020.

initial sell-off due to government stimulus measures and rapid monetary interventions by central banks. Despite the sharp drop, the S&P 500 recovered and reached new highs in the subsequent months.

Interestingly it seems that qualitatively the mean clustering and the largest eigenvalue of the stock returns increase over the last 30 years which would indicate that the market becomes more connected (largest eigenvalue is the market mode).

In Fig 3 we compute the correlation matrix of the global network properties and the market log-return using the 30 time steps (years). We observe that some variables are anti-correlated to the log return with an absolute value greater than 20%: the mean clustering and the max eigenvalue of the stock returns. This suggests that when stocks are highly connected to one another (mean clustering) or when they follow the market trend (largest eigenvalue), the global log return decreases. On the contrary, the log return is positively correlated with a coefficient greater than 20% with the community stability variable.

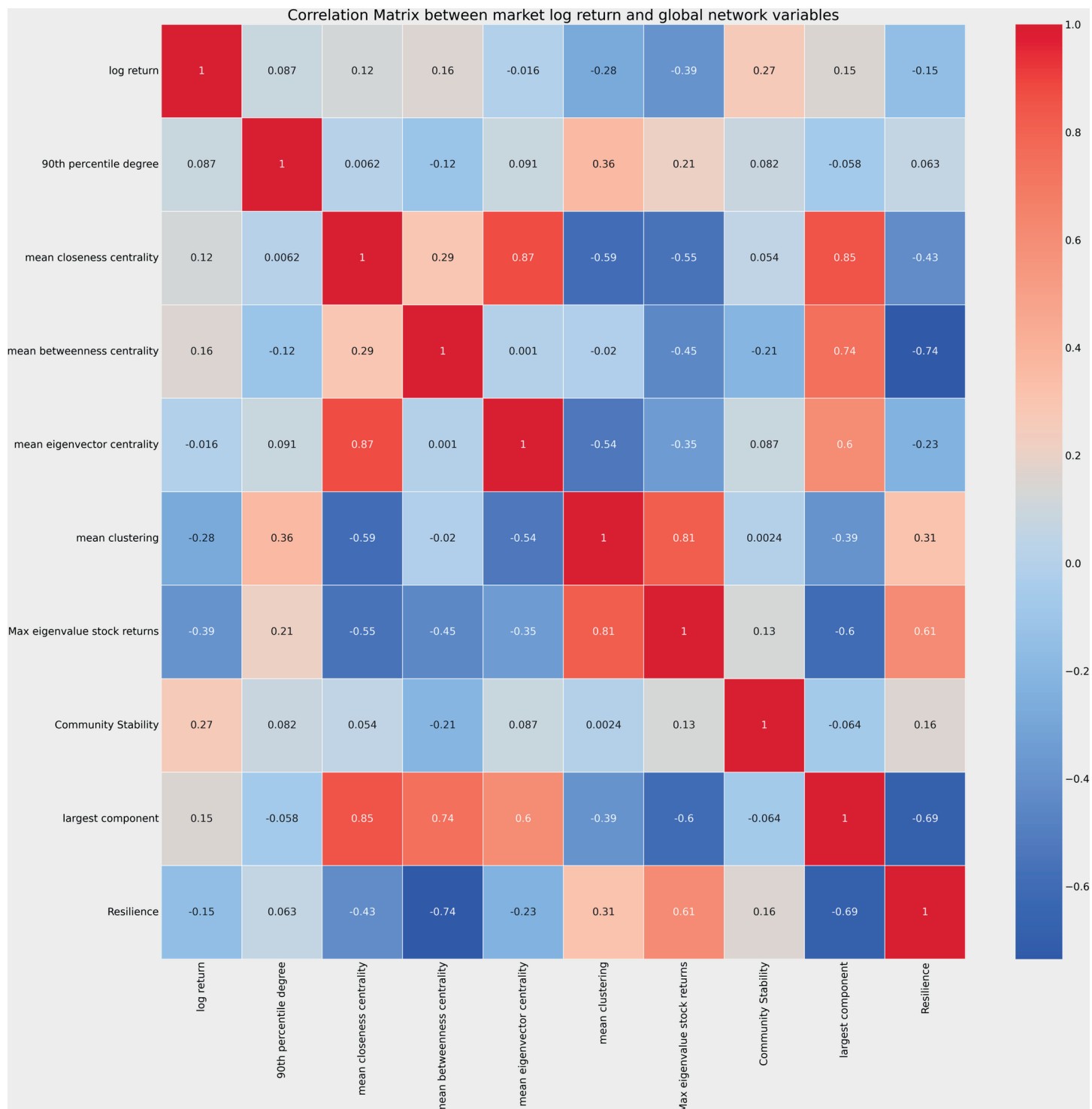

**Fig 3. Correlation matrix of the log-return of all S&P stocks with the network properties on 30 observations (in years).**

Given the limited number of observations (Nobs = 30) it is challenging to discuss the correlations between network evolution and stock returns. Nonetheless we perform a Granger causality test [56] using a maximum lag of 5 years as an indicator to test if the time series of the network properties can use to predict the stock returns. We only quote the result for the most restrictive statistical test which corresponds to the Sum of Squared Residuals (SSR) F-test for computing the $p$-value.(Other statistical tests for the SSR are $\chi^2$-test and Likelihood ratio. While we obtain $p$-values less than 0.05 in some cases, we choose to disregard them). We obtain a $p$-value less than 0.05 for the measurement given in Table 1, suggesting that these variable could be used as predictors for forecasting the global stock returns, especially the community stability at lag 1.

## The short term period

In Fig 4 we display the global dynamical evolution of the network properties (black curves) and the rescaled global stock returns smoothed with a rolling mean average window $W = 23$. The vertical dashed lines mark key market events (from left to right) [57–60]: Market Volatility Amid Inflation Concerns (Oct 2022) led to uncertainty and sharp fluctuations, SVB Bank Collapse (March 2023) triggered financial sector panic, US Debt Ceiling Agreement (June 2023) provided relief with avoided default, Tech Sector Rally (July 2023) boosted indexes driven by AI growth, Federal Reserve Interest Rate Hike (Sept 2023) slowed markets with tightened monetary policy, and Market Correction Due to Rising Bond Yields (Oct 2024) reflected investor shifts to fixed-income assets.

The matrix correlation is displayed in Fig 5, made with 238 observations (every two days). In the short time scale, the top 5 most correlated variables (in absolute value) are:

- The max eigenvalues of the stock returns, with a coefficient of –0.55, also the highest correlation on long term period.
- The mean closeness centrality, with a coefficient of 0.49
- The mean clustering, with a coefficient of –0.48, which was the second highest coefficient for the long term period.
- The largest component, with a coefficient of 0.47
- The resilience, with a coefficient of –0.35.

This is quite interesting, pointing out that the dynamical processes between the stocks on very different time scales can be partially captured by the maximum eigenvalues of the stock returns and the mean clustering of the network. The former is not surprising as it captures the market mode [14] and it was previously shown that high eigenvalues are an indicator of financial crisis [61]. The latter was also studied in [44,62,63] as a metric that can reflect the

**Table 1. Summary of significant SSR F-test $p$-values ($p < 0.05$) for Granger causality on log return with the global network variables on the long term period. The Lag represents the time delay that provides the strongest causal relationship. The 'SSR F-test $p$-value' indicates the statistical significance of the causality, with lower values suggesting stronger evidence against the null hypothesis of no causal relationship.**

| Variable | Lag | SSR F-test $p$-value |
| --- | --- | --- |
| Mean Closeness centrality | 3 | 0.0424 |
| | 4 | 0.0352 |
| Community stability | 1 | 0.0043 |
| | 2 | 0.0244 |

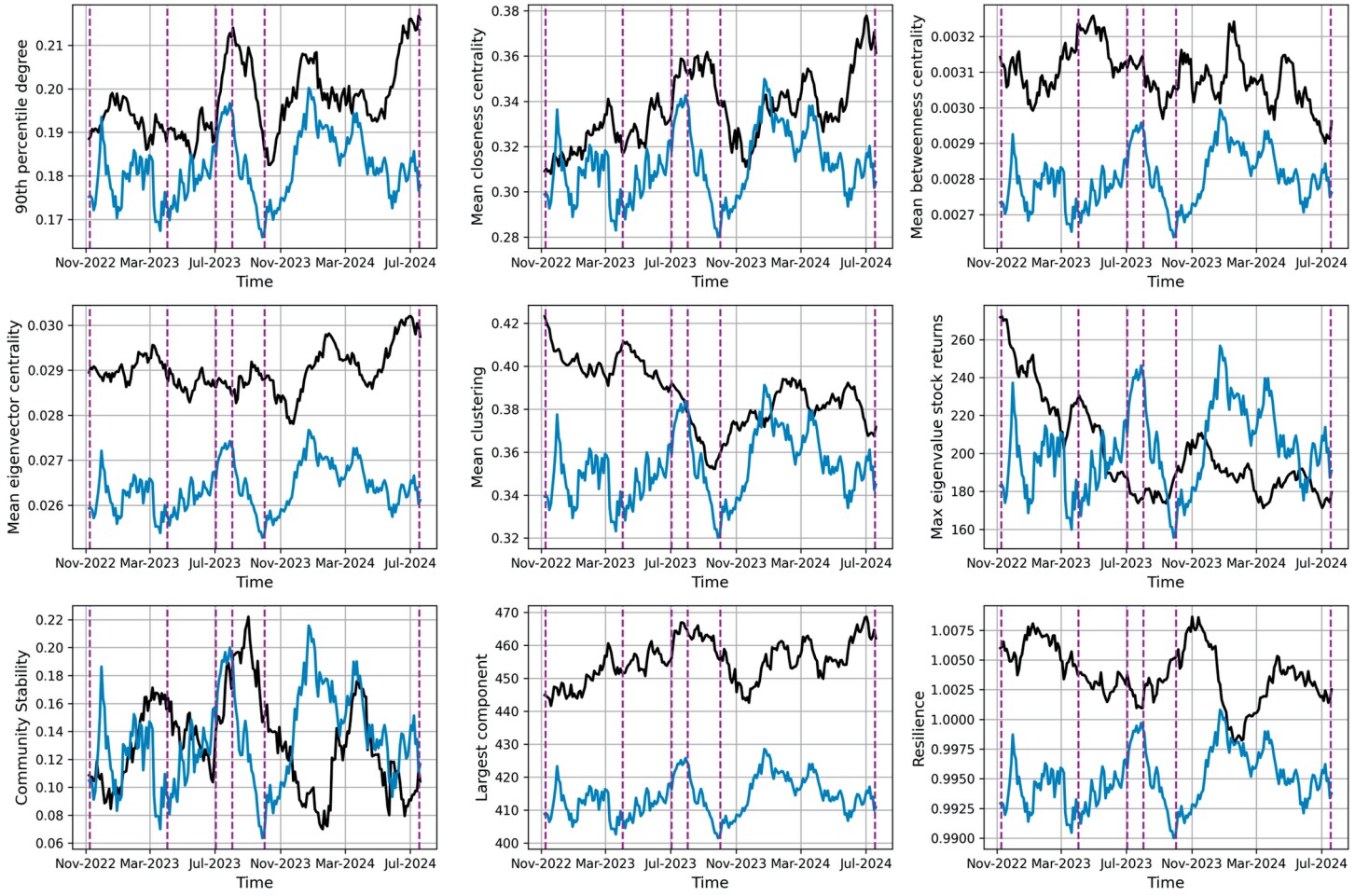

**Fig 4. Evolution of the network properties (black curves) and the stock returns rescaled (blue curves) on short time.** The short time is two business days smoothed over a rolling window $W = 23$. The number of observed time scale is *Nobs* = 238. The vertical dashed lines correspond to (from left to right): Market Volatility Amid Inflation Concerns, Oct 2022 - SVB Bank Collapse, March 2023 - US Debt Ceiling Agreement, June 2023 - Tech Sector Rally, July 2023 - Federal Reserve Interest Rate Hike, Sept 2023 - Market Correction Due to Rising Bond Yields, Oct 2024.

underlying stability of the market. Here we find that they are consistent over very different time scales when studying the global stock returns.

The Granger causality test [56] result for the best lag of our network variables is reported in Table 2, suggesting again that these variables could be used as predictors for forecasting the global stock returns. Again, it is interesting to compare these variables with the ones we find on the long-term period. For the long term period we only find the mean closeness centrality and the community stability as meaningful properties to predict the stock returns. In the short time period, we find that all 9 variables (with different lag) are correlated with the future stock return. The best *p*-value being for the Resilience at lag 3 and the 90th percentile degree at lag 2. Here the number of observation is an order of magnitude larger which makes this test more robust.

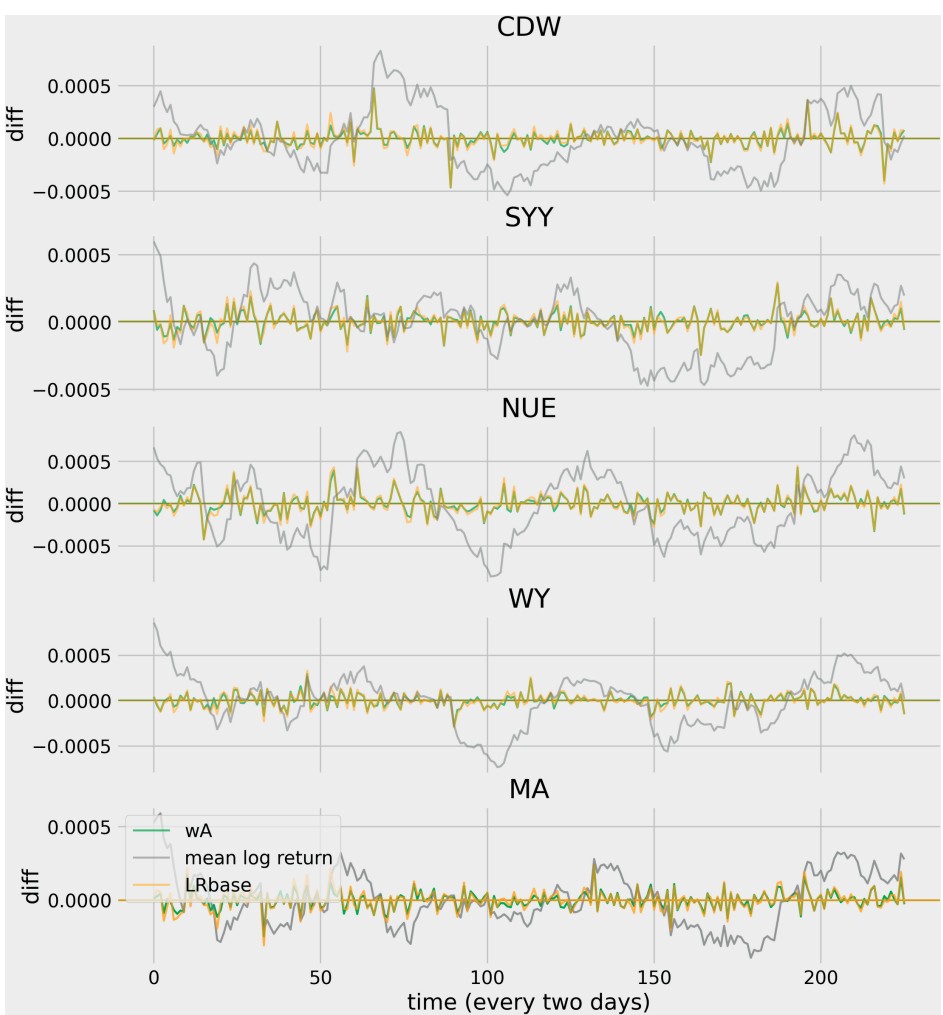

**Fig 5. Correlation matrix of the log-return of all S&P stocks with the network properties.**

**Table 2. Best Lag and SSR F-test p-value for each global network Variable on the short time period. The Best Lag represents the time delay that provides the strongest causal relationship. The 'SSR F-test *p*-value' indicates the statistical significance of the causality, with lower values suggesting stronger evidence against the null hypothesis of no causal relationship.**

| Variable | Best lag | SSR F-test *p*-value |
|---|---|---|
| 90th percentile degree | 2 | $2.16 \times 10^{-10}$ |
| Mean closeness centrality | 2 | 0.00939 |
| Mean betweenness centrality | 3 | $8.67 \times 10^{-8}$ |
| Mean eigenvector centrality | 6 | 0.00370 |
| Mean clustering | 3 | 0.00039 |
| Max eigenvalue stock returns | 1 | $4.09 \times 10^{-8}$ |
| Community stability | 1 | $4.50 \times 10^{-5}$ |
| Largest component | 2 | $5.00 \times 10^{-7}$ |
| Resilience | 3 | $1.51 \times 10^{-11}$ |

## Evolution of the individual properties of stocks within the network: Application to forecasting stock returns

Previously we have studied how the market log return correlates in time with respect to global properties of the network. We find some interesting correlations with these properties suggesting some predictability and a deeper understanding of how stocks interact with each other, leading to a coherent collective behaviour. Now we might wonder how these interactions impact individual stocks: by considering the position and properties of stock $i$ within the network, along with the global properties of the network, can we predict its log return more accurately than by using standard variables?

To address this question, we perform a forecast analysis using the following individual variables for each stock: 1- its degree centrality, 2-its closeness centrality, 3-its betweenness centrality, 4-its eigenvector centrality and 5-its clustering coefficient. These variables characterize the stock within the network. In addition we take the global network variables that we previously consider. They contain information about the overall market structure.

The standard variable that we consider for the baseline scenario is the stock's log return at the previous time step (lag).

### Methodology

**Preparing the data.** We start by randomly selecting 85% of the stocks for training and 15% for testing. For each stock we smooth all variable $f(t)$ (including the log return) using a rolling mean average of window $W$, defined as:

$$\bar{f}(t) = \frac{1}{W} \sum_{i=0}^{W-1} f(t-i) \tag{10}$$

where $\bar{f}(t)$ is the rolling mean of variable $f$ at time $t$ and $f(t-i)$ is the value of variable $f$ at time $t-i$. Note that one could adapt the average window to each variable by performing a grid search on the best forecasting results for instance. In what follows we adopt a fixed criterion for the value $W$, taking 10% of the number of the observed time.

We further compute all variables at different lag with a maximum given by Nlag. We then remove the time scale $t < Nlag$. Finally we concatenate all training stocks into one training dataset, implicitly assuming that the time series are approximately stationary.

**The regression models.** We consider multiple regression models for this analysis. For the predictions that do not use the network features but only the log-return at previous lags we use:

1- a linear regression model using solely the lag 1 of the log return as input variable (LRbase)

2- a Random Forest Regressor [64] using as input variables the most correlated lags of the log return (RFRbase)

3- an XGBoost [65] model with only the lags of the log-return (XGBbase)

4- a LightGBM [66] model with only the lags of the log-return (LGBMbase)

5- a CatBoost [67] model with only the lags of the log-return (CatBbase)

6- a mean weighted average of model 1–5 (wAbase)

For the predictions that use the network features and the log-return at previous lags we use:

1- a Random Forest Regressor using a fraction of all input variables (the most correlated ones with the lag-return defined as a Pearson correlation coefficient in the top 30th percentile of the correlation distribution) (RFR)

2- a Gradient Boosting Regressor model [68] using the same selected variables (GBR)

3- an XGBoost model on all input variables (XGB)

4- a LightGBM model on all input variables (LGBM)

5- a CatBoost model on all input variables (CatB)

6- a mean weighted average of model 1–5 (wA)

**Hyperparameters and pipelines.** For the RFRbase, RFR and GBR models we use a naive sklearn model without trying to find the optimal hyperparameters. For the XGBbase, LGBM-base, CatBbase, XGB, LGBM and CatB we perform a random sampling of the hyperparameters, use cross-validation to evaluate the performance of each hyperparameter combination and select the best model for each. The entire pipeline to reproduce our results along with the data can be accessed on https://github.com/IxandraAchitouv/Dyn_analysis_FSN_forecasting.

**Predictions and scoring.** We use the 15% of stocks (Ntest=Nstocks*0.15) not used for training and predict their log return on the entire time scale: *Nobs–Nlag*. For each stock prediction we compute the F2score [69] and the mean absolute error [70] to build their distribution of the regression models.

## Long time scale forecasting

We start with the selected S&P500 stocks from 1993-01-01 to 2024-01-01 (267 stocks) where we have Nobs = 30 (one measurement for each business year) with Nlag = 5 such that the training sample Ntrain = $(30 - 5) \times 85\% \times 267 = 5673$. We use a window $W = 10\% Nobs = 3$ to compute the rolling mean.

We perform a Granger causality test on the network variables and report the best lag *p*-value for the SSR F-test in Table 3.

The first 5 variables are specific to each stocks while the others are the global variables of the network and the max eigenvalue of the stock return correlation matrix. The result is interesting as it confirms that a stock's network properties do correlate with its future value at lag 1 but also the global properties of the network at higher lags with lower *p*-values.

The most correlated variables with the log-return (top 30th percentile of the Pearson coefficient distribution), also used for the RFR and GBR models, are reported in the appendix S1 Table. The first two most correlated variables are the log return at lag 1 and 2 followed by the

**Table 3. Best lag and SSR F-test *p*-value for each variable for the long time scale forecasting. The Best Lag represents the time delay that provides the strongest causal relationship. The 'SSR F-test *p*-value' indicates the statistical significance of the causality, with lower values suggesting stronger evidence against the null hypothesis of no causal relationship.**

| Variable | Best lag | SSR F-test *p*-value |
|---|---|---|
| Degree centrality | 1 | 0.0033 |
| Closeness centrality | 1 | $1.86 \times 10^{-15}$ |
| Betweenness centrality | 1 | $6.55 \times 10^{-5}$ |
| Eigenvector centrality | 1 | 0.02 |
| Clustering | 3 | $5.96 \times 10^{-10}$ |
| 90th percentile degree | 1 | $1.57 \times 10^{-23}$ |
| mean closeness centrality | 1 | $9.28 \times 10^{-40}$ |
| mean betweenness centrality | 4 | $7.55 \times 10^{-63}$ |
| mean eigenvector centrality | 4 | $1.1 \times 10^{-28}$ |
| mean clustering | 3 | $6.67 \times 10^{-28}$ |
| Max eigenvalue | 2 | $3.35 \times 10^{-25}$ |
| Community Stability | 3 | $3.35 \times 10^{-27}$ |
| largest component | 4 | $2.54 \times 10^{-71}$ |
| Resilience | 4 | $1.65 \times 10^{-54}$ |

90th percentile degree at lag2. The most correlated network variable that is not a global network variable is the closeness centrality at lag3. In fact over the selected network variables, 6 of them are characteristic to individual stocks and 17 describe the global network properties.

In Fig 6 we show the differences between the wA prediction and the log return (green curves), the LRbase and the log return (orange curves) and the mean of the log return with itself (grey curves) for 5 randomly selected stocks from the testing set. The horizontal lines correspond to the mean of the differences. Qualitatively, the differences between LRbase and wA are not significant but we observe that the two models do better than a simple mean average of the log return.

For each stock of the testing set we measure the R2 score and the MAE to compute their distribution for our different models. The performance summary table is given in Table 4

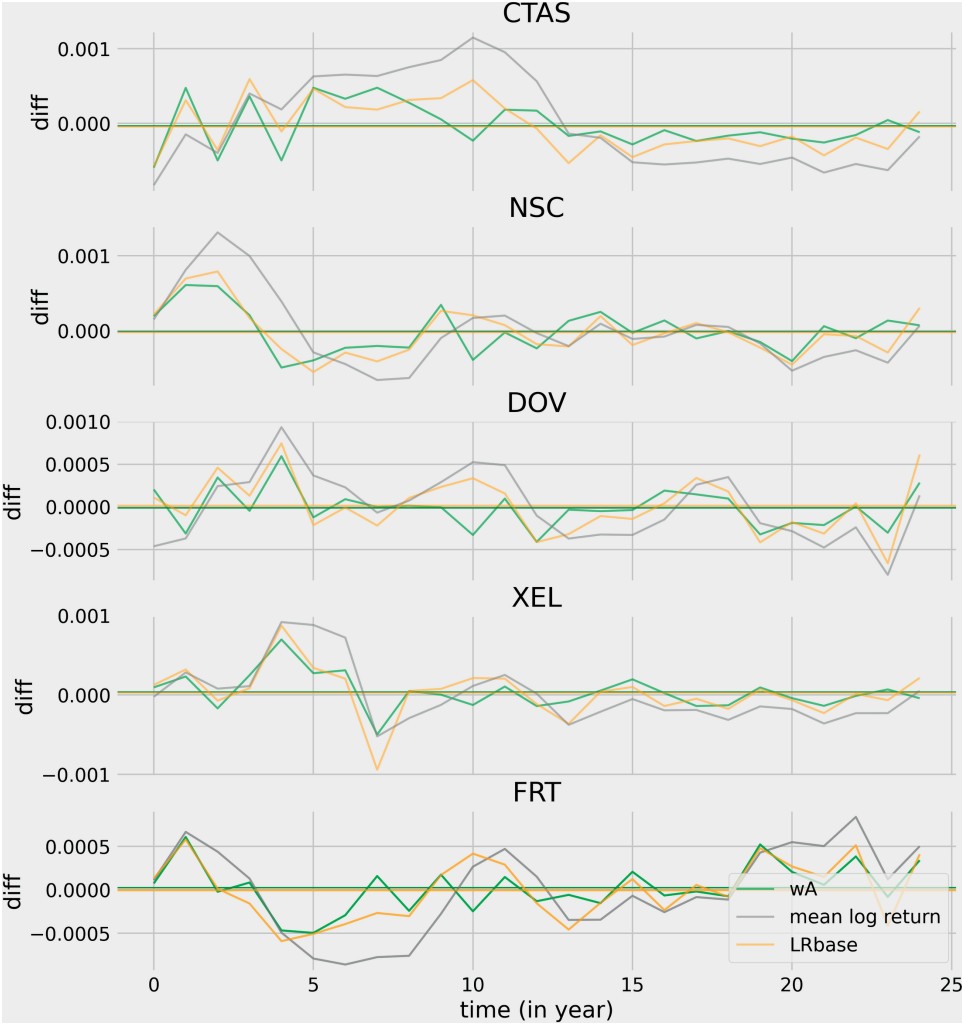

**Fig 6. Comparison of the predictions with the log return for 5 randomly selected stocks from the testing set on the long time scale.** Differences between the wA prediction and the log return (green curves), the LRbase and the log return (orange curves) and the mean of the log return with itself (grey curves). The horizontal lines correspond to the mean of the differences.

and we display in Fig 7 the distribution for some of these models. The question we want to address is whether adding the network input variables improves the overall prediction of the individual stock returns. In Table 4 we find that the relative improvement defined as $model/model_{base}$ – 1 is systematically positive, adding the network features improve all scores by an average of $\sim 21\%$ for the R2 score in average (excluding the RFR and the wA comparisons) and 22% for the MAE. Finally we observe in Fig 7 the wide skewed distribution of the R2 score and MAE point out the high variability of these simple predictive models.

## Short time scale forecasting

Similarly to the long-term forecast we use the rolling mean average window of size $W = int(10\%Nobs) = 23$ and for the number of lag we take $Nlag = 12$ after checking that including higher lag does not change the accuracy of the prediction.

Following the same methodology as described in , we report in Table 5 the Granger causality test results on the best Lag.

In the this case, the first 4 variables are associated with the individual properties of the stock within the network while the others are global properties of the network and the max eigenvalue of the stock return correlation matrix. Interestingly, the global variables have a lower $p$-value than the individual ones, suggesting that the market structure has more weight in the forecast of the individual stocks compared to the interactions of the stock within the network.

The most correlated variables with the log-return (top 30th percentile of the Pearson coefficient distribution), also used for the RFR and GBR models, are reported in the appendix S2 Table. In this case the most correlated variables are the log return itself at lag=[1,2,..12] with a correlation coefficient ranging from [0.95,0.9,...,0.44] respectively. None of the variables selected for the training are individual network variable expect for the Closeness Centrality at lag 2 and 3 that appears at rank 48 and 59 with a correlation coefficients equal to 0.09 and 0.08 suggesting again that the global network properties have more impact on the individual stock return than the stock properties within the network. Interestingly, the closeness centrality was also the most correlated individual variable in the long time period.

In the appendix we also show the differences of the predicted log return for the wA and LRbase for 5 randomly selected stocks from the testing set.

The performance summary table is given in Table 6

**Table 4. Comparison of the $R^2$ Score and Mean Absolute Error (MAE) for the long time scale forecasting.**

| Model | $R^2$ score | MAE | Relative improvement |
|---|---|---|---|
| | Median | Median | $R^2$, MAE (%) |
| RFR | 0.48 | 0.00026 | +91%, +24% |
| **RFRbase** | **0.25** | 0.00034 | |
| LRbase | 0.32 | 0.00029 | - |
| GBR | 0.48 | 0.00025 | - |
| wA | 0.58 | 0.00023 | +29%, +22% |
| **wAbase** | **0.45** | 0.00028 | |
| XGB | 0.56 | 0.00023 | +19%, +22% |
| **XGBbase** | **0.47** | 0.00028 | |
| LGBM | 0.58 | 0.00023 | +21%, +22% |
| **LGBMbase** | **0.48** | 0.00028 | |
| CatB | 0.58 | 0.00023 | +23%, +22% |
| **CatBbase** | **0.47** | 0.00028 | |

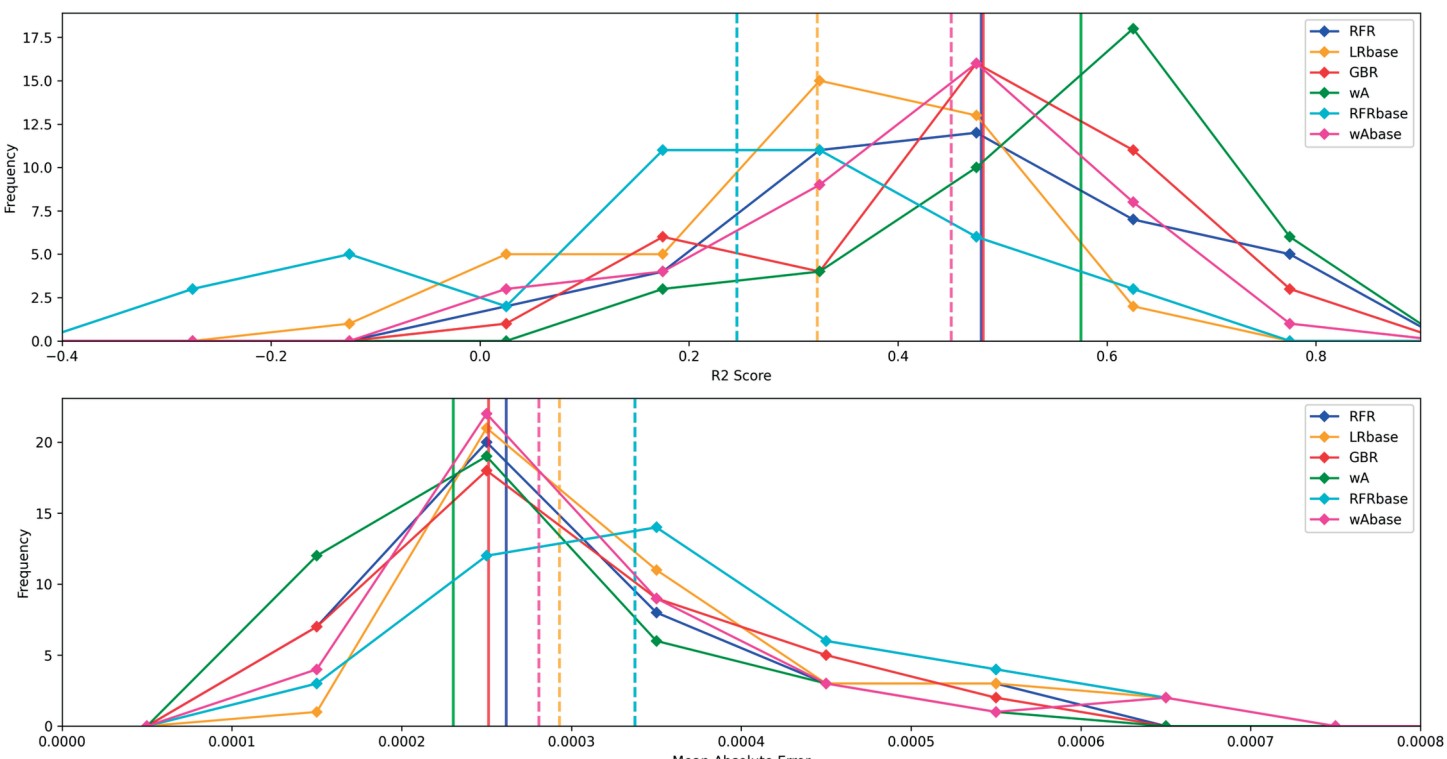

**Fig 7. Score distribution for the predicted stock returns on long time scale.** R2 distribution (top panel) and MAE distribution (lower panel) computed using the testing set of stocks for the long time scale forecasting. Vertical lines correspond to the median values. The added value of the network parameters can be observed by comparing the base models (without network features) with the others.

**Table 5. Best lag and SSR F-test *p*-value for each variable for the short time scale forecasting. The Best Lag represents the time delay that provides the strongest causal relationship. The 'SSR F-test *p*-value' indicates the statistical significance of the causality, with lower values suggesting stronger evidence against the null hypothesis of no causal relationship.**

| Variable | Best lag | SSR F-test *p*-value |
|---|---|---|
| Degree centrality | 9 | $2.43 \times 10^{-13}$ |
| Closeness centrality | 11 | $6.40 \times 10^{-37}$ |
| Eigenvector centrality | 10 | $5.12 \times 10^{-13}$ |
| Clustering | 9 | $2.49 \times 10^{-22}$ |
| 90th percentile degree | 10 | 0 |
| mean closeness centrality | 11 | $2.23 \times 10^{-111}$ |
| mean betweenness centrality | 10 | $3.22 \times 10^{-94}$ |
| mean eigenvector centrality | 10 | $3.31 \times 10^{-151}$ |
| mean clustering | 11 | $1.85 \times 10^{-154}$ |
| Max eigenvalue | 11 | $8.29 \times 10^{-186}$ |
| Community stability | 11 | $1.53 \times 10^{-232}$ |
| largest component | 8 | $6.86 \times 10^{-127}$ |
| Resilience | 11 | $6.54 \times 10^{-194}$ |

and we display in Fig 8 the distribution for some of these models. For these predictions the distribution of the R2 scores and the MAE are much better than for the long time scale. The training sample size and the higher rolling window are probably the main reasons why. Note that we also test the forecasting using different rolling average window of size *W* = 10, for

**Table 6. Comparison of the R² Score and Mean Absolute Error (MAE) for the short time scale forecasting.**

| Model | R² Score | MAE | Relative improvement |
| --- | --- | --- | --- |
| | Median | Median | R², MAE (%) |
| RFR | 0.92 | $6 \times 10^{-5}$ | +3%, +14% |
| **RFRbase** | **0.89** | $7 \times 10^{-5}$ | |
| LRbase | 0.89 | $7 \times 10^{-5}$ | - |
| GBR | 0.91 | $7 \times 10^{-5}$ | - |
| wA | 0.92 | $6 \times 10^{-5}$ | +3%, +14% |
| **wAbase** | **0.89** | $7 \times 10^{-5}$ | |
| XGB | 0.91 | $7 \times 10^{-5}$ | +2%, 0% |
| **XGBbase** | **0.89** | $7 \times 10^{-5}$ | |
| LGBM | 0.92 | $6 \times 10^{-5}$ | +3%, +14% |
| **LGBMbase** | **0.89** | $7 \times 10^{-5}$ | |
| CatB | 0.92 | $6 \times 10^{-5}$ | +3%, +14% |
| **CatBbase** | **0.89** | $7 \times 10^{-5}$ | |

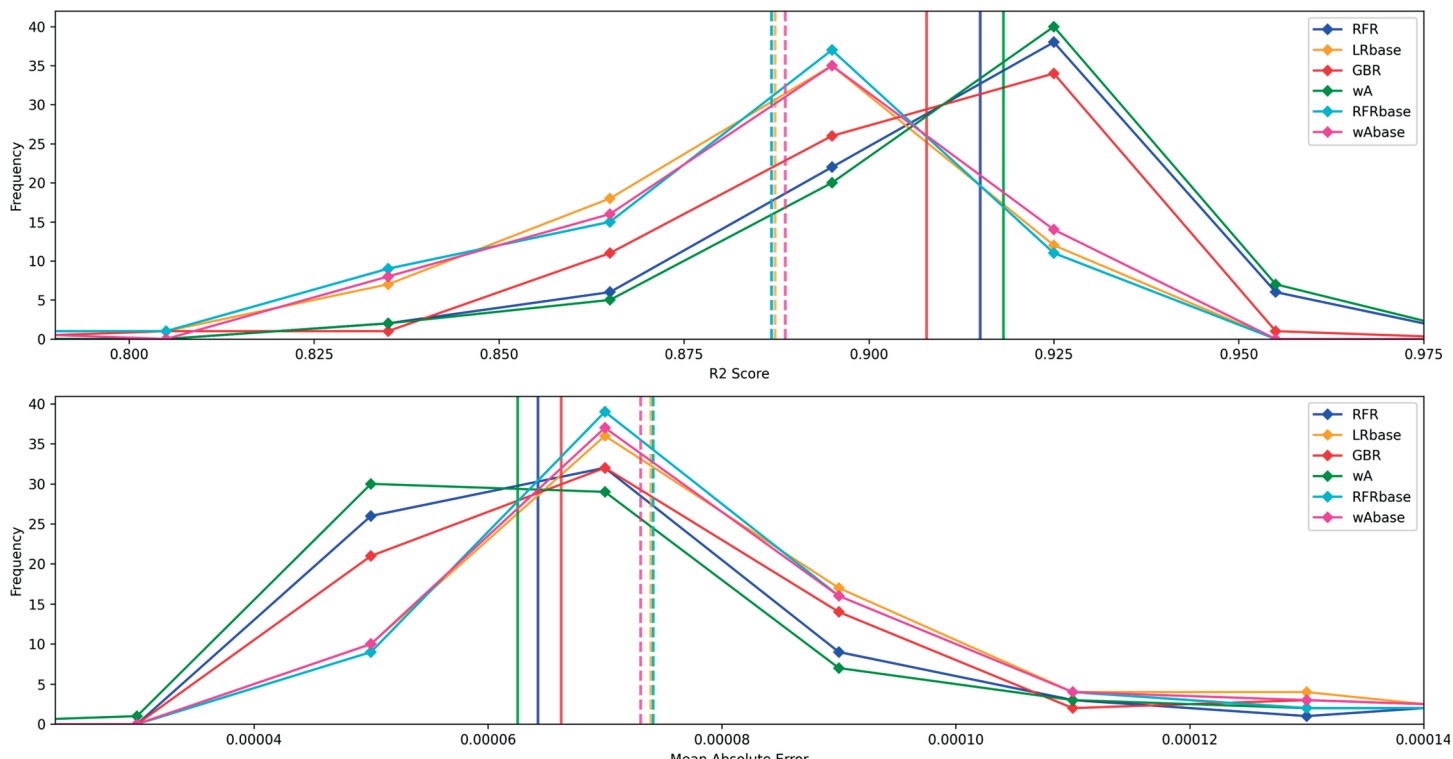

**Fig 8. Score distribution for the predicted stock returns on short time scale.** R2 distribution (top panel) and MAE distribution (lower panel) computed using the testing set of stocks for the long time scale forecasting. Vertical lines correspond to the median values. The added value of the network parameters can be observed by comparing the base models (without network features) with the others.

which the median R2score leads to a 6% improvement, for $W = 30$ we get a 2% improvement and for $W = 5$ a 21% improvement.

Nonetheless we find again that adding the network input variables improves the overall prediction of the individual stock returns. In Table 6 we find that the relative improvement defined as $model/model_{base} - 1$ is positive, adding the network features improve all scores by $\sim 3\%$ for the R2 score and 11% for the MAE, in average.

## Discussion

We find some interesting results in both the long and the short time periods. First, as expected the log return itself at previous lag is most important variable to predict the future value of a stock. Second, the global network variable are more correlated to the future return of a stock than the stock characteristics within the network. This suggests that the overall market structure plays a more important role than the stock's interactions within the network. The closeness centrality measure of a stock is the most important individual feature to predict the future return of a stock in our models. The resilience of the network and the 90th Percentile Degree are also strong global features that correlate with the future return of a stock. Overall the network variables improve the forecasting of individual stock returns both on short and long time period.

## Summary and conclusions

In this article we study the dynamical evolution of a network made from stock returns correlations and how its properties correlate with the log return itself. We consider global properties of the network as well as individual properties of the nodes. This allows us to study the correlation between 1- the market return evolution with the global properties of the network, and 2- the evolution of individual stock returns with the characteristics of the stock within the network. Both at the collective level and on the individual level we find meaningful indicators for the future evolution of the stocks (Granger causality test). We also study the dynamical evolution on two different time scales: long period and short period. We find that some indicators are scale invariant (the largest eigenvalue of the correlation matrix and the average clustering of the network) for the collective evolution. For the individual stocks, the degree of the node, the closeness centrality, the eigenvector centrality and the clustering of the node are meaningful indicators on both long- and short-term periods, although they were not selected to forecast the future value of our testing stocks for the short time period except for the closeness centrality measure. These findings are relevant for risk management as they suggest that depending on the trading strategy, such as daily or monthly trading, some network variables remain consistent across time scales. Hence, these variables could be used as indicators for measuring risk, helping investors and analysts assess market conditions and adjust strategies accordingly.

Finally, we use the network variables to forecast the evolution of stock returns and find an improvement over baseline scenarios that do not include network features (21% improvement for the long period and 3% for the short period). This indicates that network variables capture some of the interactions of the complex financial system, thus providing an added value towards predicting the dynamical evolution of collective and individual stock returns. While this study provides valuable insights, the key limitation of these finding are the reduced number of observations which could benefit from a larger dataset; however, such data were not available for this analysis. Additionally, the analysis relies on specific data sources (S&P 500), which, while appropriate for the scope of this work, could be further tested to different market stocks. Nevertheless, the results presented here are robust within the defined framework. Exploring alternative models or incorporating additional variables, such as macroeconomic indicators or extended datasets, would go beyond the scope of this article but could enhance future research in this area.

One could also add additional variables in the training dataset when predicting individual stock returns, such as the variance of some of the network variables. It would also be interesting to test some more sophisticated forecasting models.

To conclude on the policy implications of this study: the largest eigenvalue of the correlation matrix and the average clustering, could inform the development of early warning systems for market instability, offering regulators and policymakers a tool to monitor systemic risks. Similarly, the identification of meaningful indicators for individual stocks, such as closeness centrality, suggests potential applications for portfolio optimization and risk management strategies. While these implications were outside the immediate scope of this article, they highlight the potential for translating network-based analyses into practical tools for both policymakers and market participants.

The exploration of alternative machine learning models could enhance predictive accuracy and applicability. For instance, leveraging advanced methods such as graph neural networks (GNNs) or temporal convolutional networks (TCNs) could better capture the complex, dynamic interactions within financial networks. These models, combined with network variables, may offer more robust tools for predicting stock returns and systemic risks, paving the way for their integration into decision-making processes by regulators and institutional investors. While such developments are beyond the scope of this article, they represent a promising avenue for future research and policy design.

## Materials and methods

The data used for this project were downloaded from Yahoo Finance's API and can be accessed at https://github.com/IxandraAchitouv/Dyn_analysis_FSN_forecasting. The dynamical evolution of the network variables measured in this article is also publicly available at the same link, along with the ML pipeline for forecasting. The data collection and analysis for this project complied with Yahoo Finance's Terms of Service. The dataset was used solely for non-commercial academic research purposes and was processed in accordance with the platform's terms to ensure proper attribution and compliance.

## Supporting information

**S1 Fig**. Network of the stocks for different years and their properties for the short time scale. In this figure, we display the network built on the short time scale for 4 randomly selected scale [46,106,151,211] (top panels), along with the degree distribution (left bottom panels), eigenvector centrality histogram (middle bottom panels) and clustering histogram of the stocks (right bottom panels). Red vertical lines correspond to the mean. Compared to Fig 1 the number of stocks are higher. We observe some clustering per sector (shown by the colors). In average the number of clusters found with the Louvain is smaller than in the long time period.
(TIF)

**S2 Fig**. **Comparison of the predictions with the log return for 5 randomly selected stocks from the testing set on the short time scale.** Differences between the wA prediction and the log return (green curves), the LRbase and the log return (orange curves) and the mean of the log return with itself (grey curves). The horizontal lines correspond to the mean of the differences. In this fugure, we show the difference of the predicted log return for the wA, the LRbase and the mean of the log return for 5 selected stock in the testing set. There is no qualitative difference between the LRbase prediction and the wA one. Both outperform a simple mean average of the log return.
(TIF)

**S1 Table. Selected variables for the training of the long time period.**
(PDF)

**S2 Table. 42 Selected variables over 63 for the training of the short time period: we don't show the ones with a correlation coefficient less than 0.10.**
(PDF)

## Acknowledgments

I.A. would like to thank the Institut des Systèmes Complexes de Paris Île-de-France for supporting the article processing charge of this article.

## Author contributions

**Conceptualization:** Ixandra Achitouv.

**Methodology:** Ixandra Achitouv.

**Validation:** Ixandra Achitouv.

**Writing – original draft:** Ixandra Achitouv.

**Writing – review & editing:** Ixandra Achitouv.

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
