## [Decision Letter · Decision Letter 0]

26 Dec 2024

PONE-D-24-43820

Dynamical analysis of financial stocks network: improving forecasting on individual stock return using network properties.

PLOS ONE

Dear Dr. Achitouv,

Thank you for submitting your manuscript to PLOS ONE. After careful consideration, we feel that it has merit but does not fully meet PLOS ONE’s publication criteria as it currently stands. Therefore, we invite you to submit a revised version of the manuscript that addresses the points raised during the review process.

Enclosed, you will find a detailed list of observations and suggestions for your consideration.

The reviewers have highlighted several strengths in your work, such as the clear articulation of your research objectives or the novelty of applying network analysis to stock return forecasting.

At the same time, they have provided constructive feedback to address issues where further clarification on your methodology and additional analysis needed to strengthen the manuscript.

Please carefully review these comments and address them in your revised manuscript.

We appreciate if  you submit your revised article by Feb 09 2025 11:59PM. If you will need more time than this to complete your revisions, please reply to this message or contact the journal office at plosone@plos.org. Please include the following items when submitting your revised manuscript:

We look forward to receiving your revised manuscript.

Kind regards,

Alejandro Raúl Hernández-Montoya, Ph D

Academic Editor

PLOS ONE

Journal Requirements:

2. In your Methods section, please include additional information about your dataset and ensure that you have included a statement specifying whether the collection and analysis method complied with the terms and conditions for the source of the data.

5. We note you have included a table to which you do not refer in the text of your manuscript. Please ensure that you refer to Table 1-6 in your text; if accepted, production will need this reference to link the reader to the Table.

Reviewers' comments:

Reviewer's Responses to Questions

**Comments to the Author**

1. Is the manuscript technically sound, and do the data support the conclusions?

Reviewer #1: Partly

Reviewer #2: Partly

2. Has the statistical analysis been performed appropriately and rigorously? 

Reviewer #1: Yes

Reviewer #2: No

3. Have the authors made all data underlying the findings in their manuscript fully available?

Reviewer #1: Yes

Reviewer #2: Yes

4. Is the manuscript presented in an intelligible fashion and written in standard English?

Reviewer #1: Yes

Reviewer #2: Yes

5. Review Comments to the Author

Reviewer #1: Revised the manuscript according to the below comments

1. The abstract clearly conveys the study's aim and findings but could provide more specifics on implications.

2. The research objective is well-articulated, highlighting the focus on network properties for stock return forecasting.

3. The contribution paragraph effectively distinguishes this study from prior work using network analysis in finance.

4. The literature gap is well-identified, but further clarification on how it directly advances prior methodologies could enhance the context.

5. Data sources, including S&P 500 prices, are appropriate, but justification for the selected periods could strengthen reliability.

6. The methodology demonstrates rigor with network construction and statistical tests, but parameter thresholds need clearer rationale.

7. Results effectively showcase network metrics' predictive power, yet additional visual comparisons with baseline models would be beneficial.

8. The tables summarizing Granger causality tests are informative, though clearer labels and legends would aid interpretation.

9. Figures illustrating network dynamics over time are compelling but could benefit from annotations for key events.

10. The conclusion succinctly ties findings to implications but misses addressing the study's limitations explicitly.

11. Policy implications, while mentioned, are underdeveloped and could connect better to actionable insights for stakeholders.

12. Future policy directions suggest incorporating network variables but could elaborate on integrating alternative machine learning models.

13. Grammatical construction is clear throughout, with minor inconsistencies in tense needing standardization.

14. Citations are extensive and relevant, but some key references lack detailed discussion on their direct relevance to this study.

15. The data availability statement is transparent but could specify access restrictions more explicitly if any apply.

16. The discussion on scale invariance provides interesting insights but needs further elaboration on its practical implications for risk management.

17. Tables effectively summarize variable correlations, but additional columns summarizing implications would enhance their utility.

18. Overall, the manuscript contributes significantly to financial network research but could integrate broader cross-market analyses for generalization.

Reviewer #2: The paper aims at predicting stock returns by including network topological features as predictive variables. Despite the importance of the theme, the paper cannot be published in its current form. The main reason is that the prediction exercises were not performed properly. Apparently, the author applied the ML models to each stock individually, which is not usual. The values of the hyperparameters of the ML models were not reported, nor was how the hyperparameters were calibrated explained. There is no reason for the use of the wA model. The author reported some improvements brought about by the inclusion of the network variables (e.g., “In this case we have an improvement of 0.48/0.32 − 1 = 50% on the R2 score”, p. 13), but the reader cannot understand where these numbers came from. Finally, the selection of the predictive variables seems to be rather arbitrary. The author should consider all the available variables or apply some rigorous technique for feature selection. I suggest the author to i) employ a wider range of ML models across the set of stocks, ii) apply some technique for hyperparameter calibration, and iii) perform the prediction exercises with and without the network variables to observe the improvement in the performance metrics. Besides, there are other minor issues to be addressed by the author:

• Last paragraph of the introduction (p. 2): sections are not numbered (“in sec. we describe (…)”).

• Last paragraph of p. 4: I think the correct specification is Ai,j = 1 if |Ci,j| ≥ ρc, otherwise Ai,j = 0.

• The meaning of the acronyms of the economic sectors in Fig. 1 at p. 5 (AEP, DUK, etc.) is missing.

6. PLOS authors have the option to publish the peer review history of their article (what does this mean?). If published, this will include your full peer review and any attached files.

Reviewer #1: No

Reviewer #2: **Yes: **Michel Alexandre

---

## [Author Response · Author response to Decision Letter 1]

20 Jan 2025

Dear Editor,

I would like to thank you and the referees for their thorough assessments of my manuscript. I have added some essential discussions and carefully revised the manuscript, with changes highlighted in blue. Here, I present a point-by-point response to address all the referees' comments.

Reviewer #1: Revised the manuscript according to the below comments

Thank you for your careful reading and detailed feedback. I have diligently revised the manuscript based on your suggestions and incorporated additional content.

1. The abstract clearly conveys the study's aim and findings but could provide more specifics on implications.

Thank you for raising this comment. I added in the abstract: These findings highlight the potential of integrating network-based variables into stock return prediction models, which could enhance forecasting accuracy and provide a deeper understanding of market dynamics. This approach could be valuable for both investors and researchers seeking to model and predict stock behaviour in complex financial networks.

2. The research objective is well-articulated, highlighting the focus on network properties for stock return forecasting.

Thanks a lot for your comment.

3. The contribution paragraph effectively distinguishes this study from prior work using network analysis in finance.

Thanks a lot for your comment.

4. The literature gap is well-identified, but further clarification on how it directly advances prior methodologies could enhance the context.

Thank you very much for this excellent suggestion. I have added two comments in the introduction: Unlike previous works that often focuses on static network structures or market-wide influence detection, this study applies dynamic network analysis to understand how stock return interdependencies evolve over time, offering insights into the collective behaviors of the market. Previous methodologies, such as \cite{SEONG2022107608}, focused on applying network properties to predict market movements but did not investigate how these properties change over time and can influence individual stock forecasts.

And

Thus, this work extends previous studies in financial network analysis by explicitly considering the scale-invariance of network correlations over different time scales and testing how network features serve as reliable predictors for future stock returns.

5. Data sources, including S&P 500 prices, are appropriate, but justification for the selected periods could strengthen reliability.

Thanks a lot for your comment. I have added some justification in the text:

The S&P 500 index was chosen for this study as it represents a diverse cross-section of the largest U.S. companies across various sectors, making it a widely recognized benchmark for the overall stock market and an ideal candidate for studying network properties and their correlation with market dynamics. The closing stock prices can be downloaded from $Yahoo$ finance. The periods chosen for this analysis were selected to balance the availability of comprehensive data with the need to capture meaningful long-term and short-term market dynamics. For the long-term period, we selected daily stock closing prices spanning from 1993-01-01 to 2024-01-01 to ensure coverage of multiple economic cycles, including major market events such as the dot-com bubble, the 2008 financial crisis, and the COVID-19 pandemic, which provide a robust basis for analyzing the dynamics of stock return networks over extended time scales. For the short-term period, the data from 2022-08-31 to 2024-07-31 captures recent market activity, offering insights into network properties under current market conditions while focusing on higher-frequency (hourly) dynamics. Both datasets used for this analysis can be accessed on {https://github.com/IxandraAchitouv/Dyn_analysis_FSN_forecasting}

and a link to the dataset used in this analysis.

6. The methodology demonstrates rigor with network construction and statistical tests, but parameter thresholds need clearer rationale.

Thank you very much for this advice. As explained in the l 150, the value for the threshold is usually an heuristic choice as discussed in \cite{park_perspective_2020}.

To address your point and explain better how the threshold was selected in this analysis I added:

In practice, we implement a loop starting with an initial threshold value of $\rho_c = 0.8$. For each iteration, we compute the adjacency matrix and the degree distribution of the nodes. If the degree distribution is not convex, we increment the threshold by 0.1 and repeat the process. This iterative procedure continues until the degree distribution becomes convex, at which point we stop, resulting in threshold values of $\rho_c \sim 0.9$. One could see this threshold as an hyperparameter for the construction of the adjacency matrix \cite{namaki2011network}. While exploring alternative criteria for threshold selection goes beyond the scope of this analysis, we note that as long as the threshold value is sufficiently large to filter meaningful correlations, the results should remain robust.

7. Results effectively showcase network metrics' predictive power, yet additional visual comparisons with baseline models would be beneficial.

Thank you for raising this comment. In Figures 6,10 and 7,8, we compare the baseline models with the models that implement network features in predicting a few randomly selected stock evolutions, as well as in the distributions of the R² score and MAE score computed across the entire test set. To better highlight the benefits of the network, I have added the following sentence to the captions of Figures 7 and 8:

The added value of the network parameters can be observed by comparing the base models (without network features) with the others.

In addition, I have added two tables (Tables 4 and 6) that explicitly show the improvement of the network features with additional ML models and have edited the text accordingly.

8. The tables summarizing Granger causality tests are informative, though clearer labels and legends would aid interpretation.

Thank you for raising this comment. I added some explanation in the captions:

The Best Lag represents the time delay that provides the strongest causal relationship. The 'SSR F-test p-value' indicates the statistical significance of the causality, with lower values suggesting stronger evidence against the null hypothesis of no causal relationship.

9. Figures illustrating network dynamics over time are compelling but could benefit from annotations for key events.

Thank you for your useful suggestion. I have updated the Figures 2 and 4 with some key annotations in the captions and in the text:

- The vertical dashed lines correspond to (from left to right): The Asian Financial Crisis (1997), which primarily affected East Asian markets but also had a global impact. The U.S. market experienced a decline in 1998 due to global instability. - The Dotcom Bubble Burst (2000–2002) led to a sharp decline in the S\&P 500 as technology stocks crashed. - The Subprime Crisis (2008), triggered by the subprime mortgage meltdown and global banking failures, caused a significant plunge in the S\&P 500. - The Federal Reserve announced its third round of quantitative easing (QE3) in September 2012, committing to purchase 40 billion USD in mortgage-backed securities per month to boost economic growth and reduce unemployment. The S\&P 500 responded positively, gaining momentum in the latter half of 2012. - The COVID-19 pandemic caused severe market disruptions, leading to a sharp and rapid decline in the S\&P 500 in March 2020. However, the market rebounded quickly after the initial sell-off due to government stimulus measures and rapid monetary interventions by central banks. Despite the sharp drop, the S\&P 500 recovered and reached new highs in the subsequent months.

- The vertical dashed lines mark key market events (from left to right) \cite{cnbc_inflation_2022,forbes_ai_rally_2023, federal_reserve_rate_2023, nasdaq_market_correction_2024}: Market Volatility Amid Inflation Concerns (Oct 2022) led to uncertainty and sharp fluctuations, SVB Bank Collapse (March 2023) triggered financial sector panic, US Debt Ceiling Agreement (June 2023) provided relief with avoided default, Tech Sector Rally (July 2023) boosted indexes driven by AI growth, Federal Reserve Interest Rate Hike (Sept 2023) slowed markets with tightened monetary policy, and Market Correction Due to Rising Bond Yields (Oct 2024) reflected investor shifts to fixed-income assets.

10. The conclusion succinctly ties findings to implications but misses addressing the study's limitations explicitly.

Thank you for raising this point. I have added in the conclusion:

While this study provides valuable insights, the key limitation of these finding are the reduced number of observations which could benefit from a larger dataset; however, such data were not available for this analysis. Additionally, the analysis relies on specific data sources (S\&P 500), which, while appropriate for the scope of this work, could be further tested to different market stocks. Nevertheless, the results presented here are robust within the defined framework. Exploring alternative models or incorporating additional variables, such as macroeconomic indicators or extended datasets, would go beyond the scope of this article but could enhance future research in this area.

11. Policy implications, while mentioned, are underdeveloped and could connect better to actionable insights for stakeholders.

Thanks a lot for your insightful feedback. I added a paragraph to address this comment:

To conclude on the policy implications of this study: the largest eigenvalue of the correlation matrix and the average clustering, could inform the development of early warning systems for market instability, offering regulators and policymakers a tool to monitor systemic risks. Similarly, the identification of meaningful indicators for individual stocks, such as closeness centrality, suggests potential applications for portfolio optimization and risk management strategies. While these implications were outside the immediate scope of this article, they highlight the potential for translating network-based analyses into practical tools for both policymakers and market participants.

12. Future policy directions suggest incorporating network variables but could elaborate on integrating alternative machine learning models.

Thank you for suggesting to expend on this. In the main forecasting results I have added more ML regression models. I also added in the conclusion:

The exploration of alternative machine learning models could enhance predictive accuracy and applicability. For instance, leveraging advanced methods such as graph neural networks (GNNs) or temporal convolutional networks (TCNs) could better capture the complex, dynamic interactions within financial networks. These models, combined with network variables, may offer more robust tools for predicting stock returns and systemic risks, paving the way for their integration into decision-making processes by regulators and institutional investors. While such developments are beyond the scope of this article, they represent a promising avenue for future research and policy design.

13. Grammatical construction is clear throughout, with minor inconsistencies in tense needing standardization.

Thank you for your careful reading. I have replaced 'Behavior' with 'Behaviour,' 'Emergence phenomena' with 'emergent phenomena,' and made minor adjustments to commas and tense consistency.

14. Citations are extensive and relevant, but some key references lack detailed discussion on their direct relevance to this study.

Thank you for this useful suggestion. I have cited key references that, while not directly relevant to this analysis, demonstrate how the field of network analysis applied to stock returns has evolved and how complementary approaches can be used to describe complex systems such as financial markets. I believe this addition addresses your concern:

Unlike previous work that often focuses on static network structures or market-wide influence detection, this study applies dynamic network analysis to understand how stock return interdependencies evolve over time, offering insights into the collective behaviours of the market. Previous methodologies, such as \cite{SEONG2022107608}, focused on applying network properties to predict market movements but did not investigate how these properties change over time and can influence individual stock forecasts.

If that is not the case, could you point out the references you have in mind?

15. The data availability statement is transparent but could specify access restrictions more explicitly if any apply.

Thank you for raising this comment. I added a paragraph to address this comment:

The data used for this project were downloaded from Yahoo Finance's API and can be accessed at {https://github.com/IxandraAchitouv/Dyn_analysis_FSN_forecasting}. The dynamical evolution of the network variables measured in this article is also publicly available at the same link, along with the ML pipeline for forecasting.

16. The discussion on scale invariance provides interesting insights but needs further elaboration on its practical implications for risk management.

Thank you for raising this important point. I added a paragraph to address this comment:

We find that some indicators are scale invariant […]. These findings are relevant for risk management as they suggest that depending on the trading strategy, such as daily or monthly trading, some network variables remain consistent across time scales. Hence, these variables could be used as indicators for measuring risk, helping investors and analysts assess market conditions and adjust strategies accordingly.

17. Tables effectively summarize variable correlations, but additional columns summarizing implications would enhance their utility.

Thank you for raising this point. I added a paragraph to address this comment:

The correlation coefficients in Tab.\ref{tab:selecvariablelong} and Tab.\ref{tab:selecvariableshort} show interesting properties. First, we find that the log return at previous lags weighs significantly more than the network variables for the prediction of the stock's log return. Then, we see that the network variable correlations are distributed from $\sim 0.2$ to $\sim 0.1$. Interestingly, in the short time period, the Resilience variable at lags 1, 2, and 3 weighs highly in the model, as well as the maximum eigenvalue of the stock returns. For the long time period, it is the 90th percentile of the degree at lags 1 and 2 that is consistently weighted as the top network feature variable.

18. Overall, the manuscript contributes significantly to financial network research but could integrate broader cross-market analyses for generalization.

Thank you very much for your positive comment. I agree that this analysis could be extended to cross-market analysis with direct applications in business. However, this would go beyond the scope of the current study, as stated in the conclusion. This article aims to demonstrate the added value of the network as a tool for extracting signals from a complex system. The focused analysis on the S&P 500 illustrates this purpose, and I hope to have the opportunity to pursue such analyses in a broader context in the future.

Reviewer #2: The paper aims at predicting stock returns by including network topological features as predictive variables. Despite the importance of the theme, the paper cannot be published in its current form. The main reason is that the prediction exercises were not performed properly. Apparently, the author applied the ML models to each stock individually, which is not usual.

I would like to thank the referee for raising these legitimate concerns and for the opportunity to address all the points listed below, which have made the current manuscript more precise and robust in terms of ML forecasting.

I would also like to emphasize that one of the key objectives of this article is to demonstrate the added value of network variables in analyzing or predicting financial markets. The forec

---

## [Decision Letter · Decision Letter 1]

12 Feb 2025

Dynamical analysis of financial stocks network: improving forecasting using network properties

PONE-D-24-43820R1

Dear Dr. Achitouv,

We’re pleased to inform you that your manuscript has been judged scientifically suitable for publication and will be formally accepted for publication once it meets all outstanding technical requirements.

Kind regards and I sincerely apologize for the unusual delay in sending my decision on your manuscript.

Alejandro Raúl Hernández-Montoya, Ph D

Academic Editor

PLOS ONE

Additional Editor Comments (optional):

Reviewers' comments:

Reviewer's Responses to Questions

**Comments to the Author**

1. If the authors have adequately addressed your comments raised in a previous round of review and you feel that this manuscript is now acceptable for publication, you may indicate that here to bypass the “Comments to the Author” section, enter your conflict of interest statement in the “Confidential to Editor” section, and submit your "Accept" recommendation.

Reviewer #1: All comments have been addressed

Reviewer #2: All comments have been addressed

2. Is the manuscript technically sound, and do the data support the conclusions?

Reviewer #1: Partly

Reviewer #2: Yes

3. Has the statistical analysis been performed appropriately and rigorously? 

Reviewer #1: Yes

Reviewer #2: Yes

4. Have the authors made all data underlying the findings in their manuscript fully available?

Reviewer #1: Yes

Reviewer #2: Yes

5. Is the manuscript presented in an intelligible fashion and written in standard English?

Reviewer #1: Yes

Reviewer #2: Yes

6. Review Comments to the Author

Reviewer #1: I can conclude that the comments raised during the review process have been appropriately considered during the revision of the manuscript. Furthermore, the authors have taken the reviewer’s feedback seriously and addressed each point thoroughly. The current version of the manuscript meets the necessary standards and can be considered for publication in PLoS ONE. Therefore, I recommend this work for publication in its present form.

Reviewer #2: (No Response)

7. PLOS authors have the option to publish the peer review history of their article (what does this mean?). If published, this will include your full peer review and any attached files.

Reviewer #1: No

Reviewer #2: No

---

## [Editor Report · Acceptance letter]

PONE-D-24-43820R1

PLOS ONE

Dear Dr. Achitouv,

I'm pleased to inform you that your manuscript has been deemed suitable for publication in PLOS ONE. Congratulations! Your manuscript is now being handed over to our production team.

Kind regards,

on behalf of

Dr. Alejandro Raúl Hernández-Montoya

Academic Editor

PLOS ONE